# Economic burden of sickle cell disease in Brazil

**Ana Cristina Silva-Pinto[1,2], Fernando F. Costa[3], Sandra Fatima Menosi Gualandro[4], Patricia Belintani Blum Fonseca[5], Carmela Maggiuzzu Grindler[6], Homero C. R. Souza Filho[7], Carolina Tosin Bueno[7]*, Rodolfo D. Cançado[8]**

1 Regional Blood Center, Ribeirão Preto School of Medicine, University of São Paulo, São Paulo, Brazil,
2 Department of Medical Imaging, Hematology, and Oncology, Ribeirão Preto School of Medicine, University of São Paulo, São Paulo, Brazil, 3 School of Medical Sciences, University of Campinas, Campinas, SP, Brazil, 4 Department of Hematology, University of São Paulo School of Medicine, São Paulo, Brazil, 5 Department of Pediatric Hematology, Darcy Vargas Children's Hospital, São Paulo, Brazil, 6 Department of Technical Area of Neonatal, São Paulo State Health Department: Secretaria da Saude do Estado de Sao Paulo, São Paulo, Brazil, 7 Novartis Oncology, São Paulo, Brazil, 8 Department of Hematology/Oncology, Santa Casa Medical School of Sao Paulo, Sao Paulo, Brazil

* carolina.bueno@novartis.com

**Data Availability Statement:** All relevant data are within the paper and its Supporting Information files.

**Funding:** Novartis Brazil funded this study. Study design: Yes, Novartis Biociências S/A Data

## Abstract

### Background

Sickle cell disease (SCD) may cause several impacts to patients and the whole society. About 4% of the population has the sickle cell trait in Brazil, and 60,000 to 100,000 have SCD. However, despite recognizing the significant burden of disease, little is known about SCD costs.

### Objective

To estimate SCD societal costs based on disease burden modelling, under Brazilian societal perspective.

### Methods

A disease burden model was built considering the societal perspective and a one-year time horizon, including direct medical and indirect costs (morbidity and mortality). The sum of life lost and disability years was considered to estimate disability-adjusted life years (DALYs). Data from a public database (DATASUS) and the prevalence obtained from literature or medical experts were used to define complications prevalence and duration. Costs were defined using data from the Brazilian public healthcare system table of procedures and medications (SIGTAP) and the human capital method.

### Results

Annual SCD cost was 413,639,180 USD. Indirect cost accounted for the majority of burden (70.1% of the total; 290,158,365 USD *vs* 123,480,816 USD). Standard of care and chronic complications were the main source of direct costs among adults, while acute conditions were the main source among children. Vaso-occlusive crisis represented the complication

collection and analysis: Yes, Novartis Biociências S/A Decision to publish: No Preparation of the manuscript: Yes, Novartis Biociências S/A.

**Competing interests:** The authors have declared that no competing interests exist.

with the highest total cost per year in both populations, 11,400,410 USD among adults and 11,510,960 USD among children.

## Conclusions

SCD management may impose an important economic burden on Brazilian society that may reach more than 400 million USD per year.

## Introduction

Sickle cell disease (SCD) involves a group of inherited conditions in which both alleles for beta-globin are mutated and at least one is the mutation for beta-S-globin, with a quantitative predominance of hemoglobin S within the red blood cells. Depending on the mutations type, heterozygosis or homozygosis, the disease classification is sickle cell anemia (SCA; hemoglobin SS), hemoglobin SC or hemoglobin S/β-thalassaemia [1]. The disease is most commonly observed among sub-Saharan Africans; however, it is globally observed. Considering SCA frequency, it is estimated that 300,000 children are born globally with the disease per year. This number will reach 400,000 in 2050 [2].

In Brazil, a study has estimated that one in every six newborns have abnormal hemoglobins, encompassing all hemoglobinopathies [3]. Another study reported that 3.9% of the adults receiving treatment at hematology outpatient clinics have SCD [4]. Brazilian Ministry of Health estimates that about 4% of the population have the sickle cell trait and that homozygous or compound heterozygous disease is present among 60,000 to 100,000 people [5]. Furthermore, a mortality rate of 1.12/100,000 habitants is estimated [6]. Despite the efforts promoted by the Ministry of Health to propose national public policies, disparities in SCD patients' care are still observed across country regions [7].

The SCD management may include the use of hydroxyurea, folic acid, blood transfusion, iron chelators, antibiotic therapies, vaccination, and hematopoietic stem cell transplantation [5, 8]. However, only hematopoietic stem cell transplantation has a curative intention for a few patients. Hydroxyurea and red blood cell transfusion are the major disease-modifying therapies. Despite treatment, patients still experience several systemic manifestations of SCD [8, 9].

Since SCD is a multisystem disorder, every organ in the body can be affected. Patients could present multiple complications such as acute chest syndrome, acute ischemic stroke, splenic sequestration, avascular necrosis, leg ulcerations, priapism, cholelithiasis, vaso-occlusion crisis (VOCs), and others [1, 10]. A Brazilian study has reported that only 4.63% of deaths among SCD patients are unrelated to the disease. The leading causes of death arre infection (29.18%) and acute chest syndrome (25.27%) [6]. Furthermore, patients with a higher number of VOCs in a year have an increased likelihood of experiencing other SCD-related complications and death [11]. Considering that SCD patients have higher morbidity and mortality, understanding the disease burden and the impact of several complications is important to support health policy-making decisions in the country [2]. In addition, the Brazilian public health-care system has equity as one of its principles, which turns the disease burden into an impact on society [12].

The disease burden stimulated many interventions such as neonatal screening, bacterial prophylaxis, and comprehensive healthcare management [2]. In Brazil, the clinical protocol and therapeutic guidelines published by the Ministry of Health in 2018 propose similar disease management and screening [5].

Despite these initiatives and the increase in survival, mortality rates are still high in Brazil. Arduini et al. (2017) conducted a systematic review aiming to characterize mortality by SCD in Brazil in respect to the frequency, death rate or mortality coefficient, age, and causes. Mortality rates ranged from 0.115 to 0.54 per 100.000 individuals, depending on the studied population, region, and other variables. This data highlights the scarcity of information about SCD-related deaths in Brazil [13]. In addition, Brazilian SCD patients' life expectancy is about 20 years lower than the observed for the whole country [6].

Beyond the mortality rates and impact on patients' life, little is known about SCD costs. Most of the available studies reports only costs among hospitalized patients [14–17]. Kauf *et al.* (2009) reported an average total cost of care per patient-month of USD 1,946 [18]. Shah and coworkers (2020) reported mean annual cost per patient of USD 1,204 and highlighted the severe impact of VOCs on resource utilization [19]. However, only direct medical costs were included [18, 19]. In Brazil, Lobo and colleagues (2022) reported emergency visits and hospitalizations total costs higher than USD 500,000 [20]. Nevertheless, only a single study, conducted in a single center, reporting direct costs is available to date.

Considering the lack of knowledge on the SCD burden, this study aims to estimate SCD societal costs based on disease burden modelling, under a Brazilian societal perspective.

## Methods

A disease burden model was built considering the societal perspective and a one-year time horizon, including direct medical costs and indirect costs (morbidity and mortality). Although a time horizon of one year was used for years lost to disability (YLD) estimation, years of life lost (YLL) calculation used the life expectancy approach. The indirect cost estimates productivity lost due to early death.

The analysis followed the methodology proposed by Larg & Moss (2011) for a cost-of-illness evaluation. The study should include an analytical framework (study perspective and motivation, epidemiologic approach, and a well-specified question), an adequate methods definition and data regarding productivity loss and resource use (quantification, healthcare resource, and productivity loss values definition and the inclusion of intangible costs) as well as adequate data analysis and reporting (presence of a range of estimates, identification of main uncertainties, sensitivity analysis, adequate documentation and justification given for cost components, data and sources, assumptions and methods, identification of main study limitations and level of results detail to answer study questions) [21].

DALY calculation was aided by WHO's DALY calculator tool following the methodology depicted at *WHO methods and data sources for global burden of disease estimates 2000–2011* [22, 23]. Age-weighting and the discount rate were not used as the World Health Organization recommended. The definition of DALY is given by Eq 1 [24].

YLL is estimated by Eq 2, where $N_d(a)$ is the number of deaths by SCD and $L(a)$ is the life expectancy at age $a$.

YLD is given by Eq 2, through the prevalence method, where $N_p(a)$ is the number of SCD's patients at age $a$, $p(c, age\ group)$ is the prevalence of complication $c$ (can be chronic or acute) by *age group* (children or adults), $AER(c)$ is the annualized event rate of $c$, $DW(c)$ is disability weight of complication $c$, and $d(c)$ the duration of the complication $c$ from the onset until remission or death [23].

$N_d(a)$ is calculated by applying the SCD mortality rate to the prevalent population. However, the raw number from the official registry was not used as death by SCD is known to be underreported in Brazil. Instead, the resulting total number of deaths per year was distributed

through age according to the ICD-10 code D57 in the Brazilian official death registry. Thereby, the death proportion by age and the estimated total number of deaths per year were used.

Life expectancy was extracted from the National Mortality Table published by the Brazilian Institute of Geography and Statistics (IBGE) [25] and disability weights (DW) from the Global Burden of Disease 2010 (GBD 2010) and the Institute for Health Metrics and Evaluation (IHME) [26]. When DW were not found on the references mentioned above, economic evaluation literature was searched for utility values used as proxies to DW. Complications' prevalence was extracted from the literature, when available, or expert medical opinion. Annualized event rate (AER) and complication duration were calculated from DATASUS, an open database that centralizes claims from all the Brazilian public healthcare system.

Data analysis of DATASUS comprised the period from January/2008 to December/2018, considering all hospitalizations registered with ICD-10 D57 (SCD). Annualized hospitalization rates were estimated by the ratio between complications count and the total number of patients-year of DATASUS.

The average event duration was based on inpatient data related to more severe cases. Therefore, the burden of some events may be overestimated. This limitation could not be avoided as outpatient event duration data is unavailable and must be noted as a limitation of the present study.

**Eq 1. DALY definition.**

$$DALY = YLL + YLD$$

**Eq 2. YLL and YLD definition.**

$$YLL = N_d(a) * L(a)$$
$$YLD = N_p(a) * p(c, age\ group) * AER(c) * DW(c) * d(c)$$

Direct medical costs were estimated using a bottom-up strategy. A literature review identified SCD treatment and main complications. Their costing was defined through the micro-costing method, broadly defined in two steps: definition of health resource use and subsequent costing [27]. The first was extracted from published guidelines, while the latter was set from the Brazilian public healthcare system table of procedures and medications (SIGTAP). Due to many uncertainties, direct non-medical costs were not included since they can hardly be evaluated. Nevertheless, this is a conservative approach since costs, such as transportation for medical appointments, may impose an important burden, as the few reference centers are usually far from patients' residences.

Indirect medical costs were based on the average annual income in Brazil (7,416.45 USD) published by the IBGE in 2020 [28].

The total disease burden was calculated by Eq 3.

**Eq 3. Total disease burden.**

$$Indirect\ costs = DALY * Annual\ income$$
$$Direct\ costs = N_p(a) * p(c, age\ group) * AER(c) * EMC(c) + TC(age\ group)$$

Where EMC(c) is the event management cost (direct costs only) of complication *c* and TC is SCD standard management cost (hydroxyurea, folic acid, blood transfusion, iron chelators, hematopoietic stem cell transplantation–HSCT, antibiotic therapy and vaccination) by *age group*.

A literature review was performed until October 2019 using MEDLINE databases via Pubmed and Latin American and Caribbean Health Sciences Literature (LILACS). Data from

a multinational study were also used when information was not available in national literature [29]. An expert panel was further conducted, on June 17, 2020, on an online platform. Five Brazilian experts were responsible for validating all data retrieved on the literature review and provided inputs on those without published information.

Values were reported in US dollars (USD) with 1 USD = 3.88 Brazilian Reais (BRL).

Parameters' uncertainty was evaluated under a deterministic sensitivity analysis. They were varied according to their respective 95% CI, when available. Otherwise, a standard variation of ±20% was applied. The results of the analysis were expressed as a tornado diagram.

## Results

### Epidemiology

Given the information scarcity and that disease prevalence in Brazil varies in the available literature, the prevalence of 24.0 cases per 100,000 inhabitants was defined, as proposed by medical experts (ME). According to the IBGE population projection, this prevalence results in approximately 50,000 patients with the disease in 2018, segmented by age [30]. A mortality rate of 2.68/1,000,000 inhabitants was assumed, as the weighted average for men and women resulted in 558 patients lost to the disease in the same year [31]. The estimative is that the SCD mortality rate is underreported because the primary diagnosis is omitted and considered only the immediate death cause, as an acute stroke or acute chest syndrome. Total deaths were age-segmented according to mortality data from the Mortality Information System (Fig 1) [32].

### Disease complications

Table 1 shows complications' prevalence, DW extracted from literature, mean hospitalization duration of acute events, and annual incidence rate, stratified by adults and children. VOC was the most frequently observed acute complication with the longer length of hospital stay. However, the most significant disability was related to stroke. The literature reports the highest

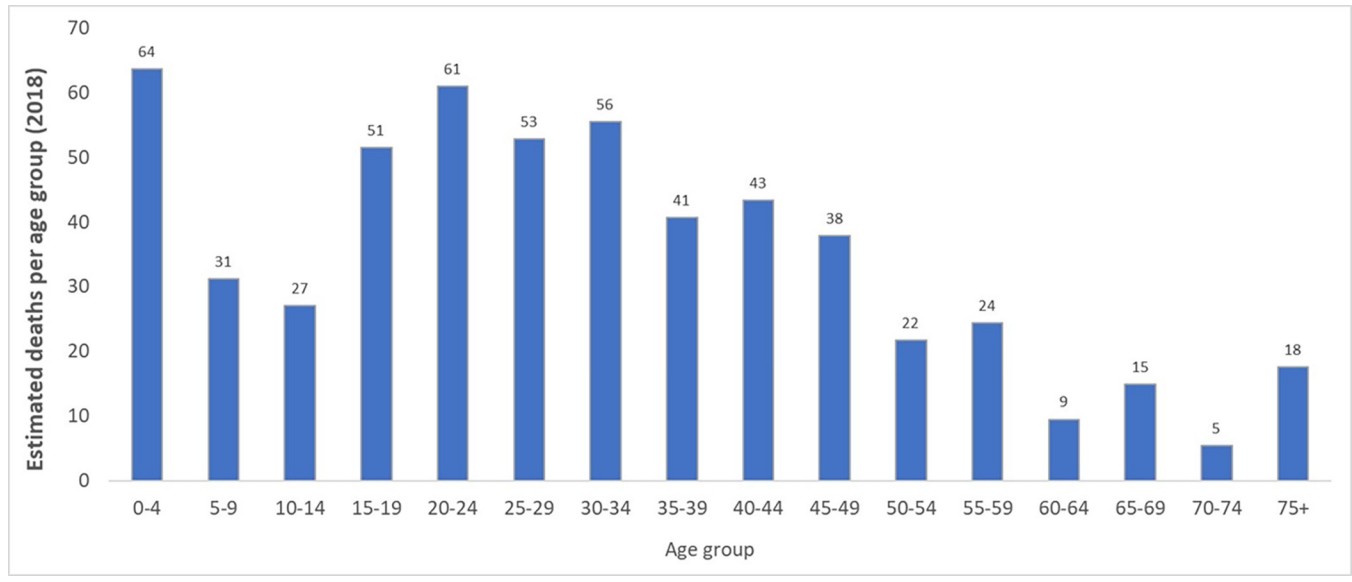

**Fig 1. Estimated number of deaths per age group among SCD patients.**

Table 1. Morbidity parameters (acute and chronic complications).

| | Adults | | Children | | Annualized hospitalization rate | Mean duration (days) | Disability Weight | |
|---|---|---|---|---|---|---|---|---|
| | % | Reference | % | Reference | | | Number | Reference |
| *Acute complications* | | | | | | | | |
| **Vaso-occlusion** | 75.0 | [33] | 59.5 | [34] | 5.30 | 15 | 0.01 | [35] |
| **Hand-Foot syndrome** | 0.0 | ME | 25.0 | ME | 5.30 | 15 | 0.01 | [35] |
| **Infections** | 32.0 | ME | 50.0 | ME | 0.12 | 5 | 0.21 | [36] |
| **Splenic sequestration** | 2.0 | ME | 34.9 | [37] | 0.09 | 6 | 0.19 | [36] |
| **Liver sequestration** | 0.1 | ME | 2.0 | ME | 0.11 | 6 | 0.18 | [35] |
| **Cholelithiasis** | 35.4 | | 27.0 | | 0.09 | 5 | 0.11 | [35] |
| **Deep venous thrombosis** | 10.0 | [38] | 2.0 | ME | 0.12 | 10 | 0.18 | [39] |
| **Acute chest syndrome** | 30.0 | ME | 55.0 | [34] | 0.11 | 7 | 0.21 | [36] |
| **Priapism** | 26.0 | [34] | 12.9 | [34] | 0.14 | 5 | 0.00 | Assumption |
| **Stroke** | 14.5 | [34] | 10.0 | ME | 0.12 | 9 | 0.43 | [35] |
| **Osteonecrosis** | 13.5 | [34] | 1.6 | [34] | 0.12 | 4 | 0.24 | [40] |
| *Chronic complications* | | | | | | | | |
| **Chronic calculous cholecystitis** | 62.0 | ME | 18.9 | ME | - | - | - | - |
| **Cardiac complications** | 15.8 | [34] | 2.6 | [34] | - | - | - | - |
| **Chronic kidney disease (failure)** | 4.0 | ME | 0.1 | ME | - | - | - | - |
| **Chronic kidney disease (without failure)** | 38.9 | [41] | 20.0 | ME | - | - | - | - |
| **Chronic liver disease** | 17.0 | ME | 8.1 | ME | - | - | - | - |
| **Leg ulcers** | 17.8 | [42] | 0.7 | [42] | - | - | - | - |
| **Osteoporosis** | 24.5 | [43] | 10.0 | ME | - | - | - | - |
| **Pulmonary hypertension** | 10.0 | [38] | 1.0 | ME | - | - | - | - |
| **Recurrent priapism** | 15.0 | ME | 6.9 | [42] | - | - | - | - |
| **Retinopathy** | 37.2 | [44] | 5.0 | ME | - | - | - | - |

ME: Medical Expert.

frequency for chronic calculous cholecystitis and chronic kidney disease (without failure) as chronic conditions among adults and children, respectively.

## Direct medical costs

Table 2 shows the standard of care costs per patient. Annual costs attributed to chronic and acute complications are shown in Table 3. SCD related cost was composed by the summation of the standard of care and chronic complications' costs. Acute complications and HSCT costs were calculated per event, not as chronic costs. The standard of care average costs assumed were 1,835.92 USD and 987.21 USD, while for chronic complications were 769.30 USD and 116.62 USD, for adults and children, respectively. Acute complications costs were estimated as 595.68 USD and 703.71 USD for adults and children, respectively. Despite its high cost per event, HSCT-related costs were estimated as 60.94 USD and 39.61 USD for adults and children, respectively. This apparent inconsistency is directly related to the low incidence of the procedure.

## Economic burden

Annual SCD cost in Brazil was approximately 414 million USD or 1.6 billion BRL per year, of which 290 million USD (1.1 billion BRL) and 123 million USD (479 million BRL) were related to indirect and direct costs, respectively. Approximately 41,000 DALYs were lost in 2018, 27,000 due to death, and 14,000 due to disability (Table 4).

**Table 2. Standard of care cost.**

| | Adults | | Children | | Cost per year adults (USD) | Cost per year children (USD) | Cost reference |
|---|---|---|---|---|---|---|---|
| | % | Reference | % | Reference | | | |
| **Medical visit** | 100 | ME | 100 | ME | 7.73 | 10.31 | SIGTAP |
| **Hydroyurea** | 50.0 | ME | 29.0 | [42] | 282.22 | 98.21 | SIGTAP |
| **Folic acid** | 96.5 | ME | 97.3 | ME | 1.36 | 1.37 | BPS (01/2020) |
| **Blood transfusion** | 11.0 | ASH 2017 | 10.6 | [42] | 192.03 | 185.05 | Estimated |
| **Iron chelators** | 15.0 | ME | 12.0 | [42] | 1352.58 | 529.76 | BPS (06/2020) |
| **Antibiotic prophylaxis** | 0.0 | ME | 15.0 | ME | - | 79.46 | BPS (01/2020) |
| **Vaccine** | 0.0 | ME | 87.4 | ME | - | 83.05* | PAHO (2020) |
| **Total cost per patient (BRL)** | - | | - | | 1,835.92 | 987.21 | - |

BPS: *Banco de preços em saúde;* USD: American dollars; ME: Medical Expert; PAHO: Pan-American Health Organization; SIGTAP: Brazilian public healthcare system table of procedures and medications.

* Pneumococcal polysaccharide vaccine 23-valent / 10-valent pneumococcal conjugate vaccine (PCV-10).

**Table 3. Chronic and acute complications health-state costs.**

| | Adults | | Children | | Cost per year/event (USD) |
|---|---|---|---|---|---|
| | % | Reference | % | Reference | |
| | *Acute complications* | | | | |
| **Vaso-occlusion** | 75.0 | [33] | 59.5 | [34] | 130.36 |
| **Hand-foot syndrome** | 0.0 | ME | 25.0 | ME | 130.36 |
| **Infections** | 32.0 | ME | 50.0 | ME | 993.43 |
| **Splenic sequestration** | 2.0 | ME | 34.9 | [37] | 533.58 |
| **Liver sequestration** | 0.1 | ME | 2.0 | ME | 314.57 |
| **Cholelithiasis** | 35.4 | [42] | 27.0 | [45] | 164.31 |
| **Deep venous thrombosis** | 10.0 | [38] | 2.0 | ME | 238.95 |
| **Acute thoracic syndrome** | 30.0 | ME | 55.0 | [34] | 529.44 |
| **Priapism** | 26.0 | [34] | 12.9 | [34] | 115.70 |
| **Stroke** | 14.5 | [34] | 10.0 | ME | 434.07 |
| **Osteonecrosis** | 13.5 | [34] | 1.6 | [34] | 110.39 |
| **Health state cost (USD)** | 595.68 | | 703.71 | | - |
| | *Chronic complications* | | | | |
| **Chronic calculous cholecystitis** | 62.0 | SWAY | 18,9 | SWAY | 147.19 |
| **Cardiac complications** | 15.8 | [34] | 2,6 | [34] | 254.05 |
| **Chronic kidney disease (failure)** | 4.0 | ME | 0,1 | ME | 8,398.20 |
| **Chronic kidney disease (without failure)** | 38.9 | [41] | 20,0 | ME | 113.54 |
| **Chronic liver disease** | 17.0 | ME | 8,1 | ME | 103.65 |
| **Leg ulcers** | 17.8 | [42] | 0,7 | [42] | 611.28 |
| **Osteoporosis** | 24.5 | [43] | 10,0 | ME | 157.66 |
| **Pulmonary hypertension** | 10.0 | [38] | 1,0 | ME | 267.40 |
| **Recurrent priapism** | 15.0 | ME | 6,9 | [42] | 227.30 |
| **Retinopathy** | 37.2 | [44] | 5,0 | ME | 85.80 |
| **Health-state cost (USD)** | 769.30 | | 116.62 | | - |
| **HSCT** | 0.2 | ME | 0.1 | DATASUS | 28,364.31 |
| **Follow-up** | 100 | ME | 100 | ME | 2,105.50 |
| **Health-state cost (USD)** | 60.94 | | 39.61 | | - |

USD: American dollars; HSCT: hematopoietic stem cell transplantation; ME: Medical Expert; SWAY: Sickle Cell World Assessment Survey.

**Table 4. Disease burden.**

| Outcome | Adults | Children |
|---|---|---|
| *Direct medical costs (USD)* | | |
| **Standard of care** | 40,390,236 | 27,642,008 |
| **Chronic complications** | 16,924,650 | 3,265,344 |
| Chronic calculous cholecystitis | 2,007,717 | 778,947 |
| Chronic kidney disease (without failure) | 971,686 | 635,831 |
| Retinopathy | 702,177 | 120,118 |
| Osteoporosis | 849,807 | 441,458 |
| Leg ulcers | 2,393,783 | 119,811 |
| Chronic liver disease | 387,639 | 235,071 |
| Cardiac complications | 883,063 | 184,945 |
| Recurrent priapism | 750,086 | 439,141 |
| Pulmonary hypertension | 588,278 | 74,872 |
| Chronic kidney disease (failure) | 7,390,414 | 235,150 |
| **Acute complications** | 13,105,040 | 19,703,765 |
| Vaso-occlusion | 11,400,410 | 11,510,960 |
| Hand-foot syndrome | 0 | 4,836,538 |
| Infections | 824,297 | 1,639,227 |
| Splenic sequestration | 21,338 | 473,897 |
| Liver sequestration | 738 | 18,790 |
| Cholelithiasis | 120,495 | 116,968 |
| Deep venous thrombosis | 63,408 | 16,140 |
| Acute thoracic syndrome | 378,095 | 882,221 |
| Priapism | 92,231 | 58,241 |
| Stroke | 165,094 | 144,910 |
| Osteonecrosis | 38,933 | 5,873 |
| **HSCT** | 1,340,671 | 1,109,101 |
| **Total cost** | 123,480,816 | |
| *Indirect costs* | | |
| **DALY** | 22,691 | 18,044 |
| YLD | 5,984 | 7,627 |
| YLL | 16,706 | 10,417 |
| **Indirect costs (USD)** | 168,284,496 | 121,873,869 |
| **Total cost (USD)** | 290,158,365 | |
| *Direct medical costs + indirect costs* | | |
| **Total cost (USD)** | 413,639,180 | |

USD: American dollars; DALY: Disability-adjusted life years; HSCT: hematopoietic stem cell transplantation; YLD: Years lived with disability; YLL: Years of life lost.

## Deterministic sensitivity analysis

Parameters that most influenced the model results were: SCD prevalence, annual income, VOC annualized rate, and SCD base utility. The first three parameters increment resulted in an increased disease burden. At the same time, the SCD base utility improvement reduced the disease burden. This behavior is expected once the improvement of SCD base utility results in reduced patient absenteeism. Fig 2 shows the deterministic sensitivity analysis complete results. Values and ranges considered for each parameter are in the S1 File.

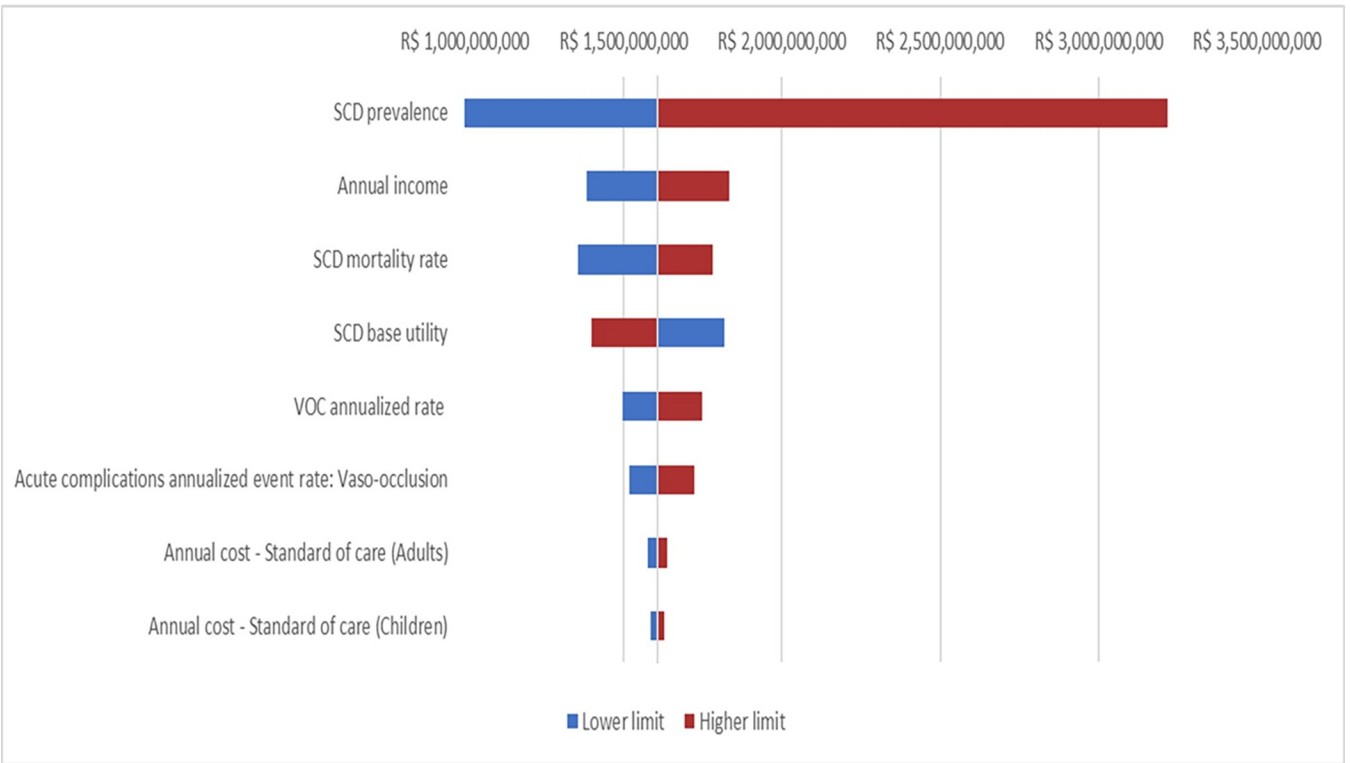

**Fig 2. Tornado diagram assessing each parameter role in model results.**

## Discussion

This study aimed to estimate the SCD burden, considering the Brazilian societal perspective. To the best of the authors' knowledge, this is the first study to assess Brazil's SCD burden and add important knowledge to improve disease management and decision-making.

Economic burden related to several hematologic conditions were previously conducted in Brazil [20, 46–50]. However, there are no studies using similar methods to estimate burden of illness in the country to date.

Data related to disease epidemiology were collected to allow SCD burden estimation in Brazil. Data available at the information system was analyzed stratified by age groups regarding mortality. It was possible to notice that mortality rates are still higher in young ages, such as children. This pattern is quite different from those observed in developed countries. Payne *et al.* (2020) assessed the trends in SCD-related mortality among black Americans considering 1979–2017 period and reported a decline in death rates among children and an increase among adults, with the median age at death increasing from 28 to 43 years [51]. Thus, it is possible to highlight that the disease burden may still be higher in developing countries, such as Brazil, considering the delay in disease management. High mortality rates were also reported in other studies. Cançado et al. (2021) reported that SCD is related to a reduction of approximately 37 years on median age at death, compared to the general population, which was also observed when data were stratified by age [52].

In the present study, the total cost related to SCD in a year was estimated at 413,639,180 USD, considering both adults and children. Most of this burden was related to indirect costs, representing 70.2% of the total amount. Previous studies worldwide assessing SCD burden reported, in the majority, data considering only those related to direct medical costs [53–61].

Naik *et al.* (2019) reported a total direct annual cost that ranged from 1 million USD to 3 million USD and an average annual indirect cost of 1,293 USD in a systematic review aiming to synthesize worldwide SCD economic burden. The lack of detailed information on Naik *et al.* (2019) does not allow us to understand the reason for the differences observed [62]. However, in another review that aimed to estimate the SCD economic burden in the United States, the authors highlighted the need for studies considering direct and indirect costs to characterize the full burden of disease [61]. Thus, despite the differences among estimates shown in the present study and those in the analysis from Naik *et al.* (2019), the impact of indirect costs on total SCD burden still needs to be better understood in further studies [62].

Still, regarding the indirect disease burden, the estimated total DALY loss in 2018 was 22,755 and 18,085 among adults and children, respectively. Considering both premature death and the years lived with disability, the disease substantially impacts the whole Brazilian society. Similarly, Rezaei *et al.* (2015) assessed DALYs estimations due to hemoglobinopathies (thalassemia, SCD, and G6PD-D) in Iran by sex and age in 1990, 2005, and 2010. Using data from Global Disease Burden, the study reported total DALY among SCD patients of 51,129 and 30,501 in 1990 and 2010, respectively, showing a decreasing trend across the years. However, in contrast to our findings, DALYs lost per adult were lower than those lost per children (369 among children ≤14 years *vs.* 88 among adults ≥15 years, in 1990; 204 among children ≤14 years *vs.* 66 among adults ≥15 years, in 2010) [63].

The SCD direct medical costs were estimated at 123,480,816 USD for 2018 in Brazil. Standard of care was the main driver of direct costs, followed by acute complications in both populations. However, among children, acute complications costs were about six times higher than those observed for chronic ones, while among adults, values were closer. This difference is influenced by the frequency of such complications since children have a higher frequency of acute conditions and adults the chronic ones [34, 42].

Considering total costs related to complications' management in a year, VOC represents the most expensive acute condition in both populations. About 87% of all acute complications' costs were related to VOCs among adults.

The condition's occurrence may explain this finding since VOC was the most frequently observed complication among children (59.5%) and adults (75.0%). In addition, the condition was responsible for the highest annualized hospitalization rate (5.30 per year) and length of hospital stay (mean duration of 15 days). Previous studies in Brazil reported that acute painful episodes are the leading cause of hospitalization among SCD patients, which may reach about 70% of the cases [64, 65].

VOCs disrupt blood circulation, which drives acute and chronic pain beyond the damage to key organs such as the liver, brain, lungs, and kidneys [66]. In addition, the condition is directly associated with mortality, with a risk of death 5.5 times higher among patients with ≥3 VOCs per year compared to those with <1 episode [11]. In the present analysis, the estimated cost per event was 130.36 USD. Despite the low cost, SCD patients may experience ≥ 6 VOCs per year, which explains the amount spent in a year (11,400,410 USD among adults and 11,510,960 USD among children) [67]. The burden related to VOC may still be higher since some issues like the hospital beds occupation were not considered in this analysis. These data highlight the importance of strategies to control disease to avoid VOCs occurrence. Furthermore, patients with uncontrolled disease seems to have a higher resource utilization and costs [68].

This study was designed to follow the checklist proposed by Larg & Moss (2011), and most of the items were addressed. However, it was impossible to control confounders and provide the required level of detail of resource use and productivity loss. Data were extracted from literature and reports from key-opinion leaders, which may not represent the Brazilian scenario in some

cases. Still, it was the available source when the study was conducted. A deterministic sensitivity analysis was conducted to estimate the influence of the model parameters on the modeled results to address some of these limitations. It is important to note that parameters were varied according to their respective 95% CI, when available, or using a standard variation of ±20%, considered wide enough to evaluate the model behavior under uncertainty [21]. The results highlight the model's sensitivity, especially to SCD prevalence, which is almost natural as the increase in the number of cases will inevitably lead to a higher burden of the disease. The same will happen to an increase in annual income as the magnitude of the results is directly related to indirect costs, income dependent. Finally, an increase in SCD base utility will reduce DW (better quality of life), reducing the disease burden. Somehow, deterministic sensitivity analysis results give rise to the necessity of SCD prevalence studies, which can refine future disease burden analysis.

Thus, although the study adds valuable knowledge, these limitations need to be highlighted. In addition, the absence of direct non-medical costs may translate to an underestimation of disease burden since the availability of reference centers in a few places may represent the need for frequent transportation for medical assistance, change of residence place, and others. However, in contrast, mortality quoted comes from a specialist hospital and may translate into a more severe population. Considering that SCD patients who are well may not need such resources, further analysis addressing this limitation is needed. Finally, the hospitalization rate was estimated through data available from DATASUS. Although it is a national source of information, only those related to public assistance are reported, and it is usually not registered as a condition related to SCD, underestimating the occurrence of the outcome.

## Conclusion

The present analysis showed that SCD patients might generate an economic burden for the Brazilian society of approximately 400 million USD per year. The indirect burden is responsible for most of this cost; however, disease complications play an important role in direct medical costs. Thus, the present data highlights the importance of investment in disease control to reduce its impact on patients and society.

## Supporting information

**S1 File.**
(DOCX)

## Acknowledgments

We thank SENSE Company for its support with medical writing during the development of this manuscript.

## Author Contributions

**Writing – original draft:** Ana Cristina Silva-Pinto, Fernando F. Costa, Sandra Fatima Menosi Gualandro, Patricia Belintani Blum Fonseca, Carmela Maggiuzzu Grindler, Homero C. R. Souza Filho, Carolina Tosin Bueno, Rodolfo D. Cançado.

**Writing – review & editing:** Ana Cristina Silva-Pinto, Fernando F. Costa, Sandra Fatima Menosi Gualandro, Patricia Belintani Blum Fonseca, Carmela Maggiuzzu Grindler, Homero C. R. Souza Filho, Carolina Tosin Bueno, Rodolfo D. Cançado.

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
