## [Decision Letter · Decision Letter 0]

11 Feb 2022

PONE-D-22-01357Economic burden of sickle cell disease in BrazilPLOS ONE

Dear Dr. Bueno,

Thank you for submitting your manuscript to PLOS ONE. After careful consideration, we feel that it has merit but does not fully meet PLOS ONE’s publication criteria as it currently stands. Therefore, we invite you to submit a revised version of the manuscript that addresses the points raised during the review process.

We look forward to receiving your revised manuscript.

Kind regards,

Mohamed A Yassin, MD

Academic Editor

PLOS ONE

Journal Requirements:

2.Thank you for stating the following financial disclosure: 

"Novartis Brazil funded this study."

Reviewers' comments:

Reviewer's Responses to Questions

**Comments to the Author**

1. Is the manuscript technically sound, and do the data support the conclusions?

Reviewer #1: Partly

Reviewer #2: Yes

2. Has the statistical analysis been performed appropriately and rigorously? 

Reviewer #1: I Don't Know

Reviewer #2: Yes

3. Have the authors made all data underlying the findings in their manuscript fully available?

Reviewer #1: Yes

Reviewer #2: Yes

4. Is the manuscript presented in an intelligible fashion and written in standard English?

Reviewer #1: No

Reviewer #2: Yes

5. Review Comments to the Author

Reviewer #1: PONE-D-22-01357Economic burden of sickle cell disease in Brazil

The authors use publicly available data to determine the economic burden to the society of providing medical care for people living with SCD (PLWSCD). They include indirect costs, an important component of total economic cost.

Abstract

The English can be improved. For example, “Sickle cell disease (SCD) may cause several impacts to patients and the whole society.”

The use of the BRL is difficult for the international audience. Consider using a commonly understood currency equivalent such as the USD or GBP, or at least stating whet he exchange rate was in the captioned year.

Introduction

The English can be improved. While it is difficult to give specific examples, the manuscript would benefit from thorough review by a proofreader. Some examples include “composed heterozygous disease”. “Furthermore, a mortality rate of 1.12/100,000 habitants is estimated”, “Only hematopoietic stem cell transplantation has a curative intention, however, for few patients.”, “disparities on SCD patients’ care”, “It was possible to notice that mortality rates are still higher in young ages, such as children.”.

The mortality quoted comes from a specialist hospital “Furthermore, a mortality rate of 1.12/100,000 habitants is estimated [6]”. It has been shown that this is a biased (more severe) population and that ideally cohorts assembled at newborn screening are the best approach. Given that “In Brazilian context, direct non-medical costs, such as transport for medical appointments, may impose important burden since disease reference sites are located in few places.”, PLWSCD who are well may not make the trip. This limitation should be acknowledged.

The authors say that “Considering the disease burden, many interventions such as neonatal screening, bacterial prophylaxis and a comprehensive healthcare management have been established [2]”, quoting an international commentary and not speaking to how often these interventions are applied in Brazil.

Similarly, the cost of care quoted was from Florida. “Kauf et al. (2009) reported an average total cost of care per patient-month of USD1,946, however only direct medical costs were included [16].”

The population of Brazil is approximately 212.6 million. The upper limit of the estimated SCD population is 100,000. This gives a maximum estimate of prevalence of 0.5% of the total population having SCD. The fact that people living with SCD (PLSCD) have related deaths and morbidity does not clearly show that “As a consequence, SCD patients have greater morbidity and mortality which could impact the whole society”.

There should be a reference at the end of this critical sentence “The analysis followed the methodology proposed by Larg & Moss (2011) for a cost-of-illness evaluation”.

Agree that reference is missing “Direct medical costs were estimated using a bottom-up strategy with the identification of the main health states related to SCD (Error! Reference source not found.)”.

There are references published before October 2019 which have been omitted. Arduini (2017) looks at several studies regarding SCD mortality in Brazil including one by Ramos (2015) from which he quotes an estimate of a mortality coefficient of 0.54 per 100,000 individuals.

Figures 1 and 2 are unhelpful. It is not clear what they are saying.

Methods

Given that it is now 2022, the halting of the literature review more than two years ago is unacceptable. There have been significant publications since then which must be mentioned in discussing how use of the different estimates may have affected the final outcomes. “Literature review was performed until October 2019”. Santo et al (2020) estimate annual mortality rates at approximately 2.65 and 2.7 per million for men and women respectively. This is very different from the 1.1% used here and would impact estimates significantly. Sensitivity analyses for these possibilities would be necessary. The YLL would be significantly lower using these estimates.

What does this sentence mean? “Disability attributed to chronic complications was incorporated from the sickle cell anemia DW”.

Results

It is not clear where the following sentence comes from. Is it a result arising from this research? “A prevalence of 24.0 cases per 100,000 inhabitants was considered, resulting in approximately 50,000 patients with the disease in 2018, segmented by age according to the IBGE population projection [24].”

Similarly, is the calculation all done on the opinion of the medical expert? Most of the calculated costs come from indirect costs which are based on these premises.

“Considering the scarcity of information and that disease prevalence in Brazil varies in the available literature, data was defined as an assumption proposed by ME. A disease specific mortality rate of 1.12% was assumed, resulting in 560 patients lost to the disease in the same year [6].

The quoted study is not the only publication on SCD mortality in Brazil.

Table1. For acute complications, please clarify whether the presented %s are for the captioned year. Did 26% of all PLWSCD, and therefore approximately 52% of all men have priapism annually? Did 35.4% of PLWSCD have cholelithiasis annually;14.4% stroke, 32% infections (not including ACS)?

The opinions of medical experts can be influenced by their type of practice. Those in secondary or tertiary care may not see the asymptomatic persons. An example is proliferative sickle retinopathy. The reference (17). The estimate is taken from a single, cross-sectional study and ME. The estimate is much higher than that in a cohort study.

Is Figure 3 Are the deaths shown from SCD or the general population?

Are the YLL based on those who died only in the year under review?

More details are needed on how disability was assigned. I do not understand how the DW were calculated. The DW do not make intuitive sense to me. Stroke has DW of 0.08, but this acute episode has chronic disability associated. ACS which usually has no disability after the episode has a DW of 0.33.

How is the indirect cost calculated? In the last part of Table 4 (see below), the direct cost is $479,105,564. Is the DALY not the source of the indirect costs? By what calculation is the DALY 22,755 transformed to the indirect cost 654,790,110

Figure 4. This needs a proper title stating what Tornado diagram depicts.

What does SCD prevalence represent in terms of costs in the Tornado diagram?

Discussion

The high death rate among children is surprising, given that there is newborn screening, vaccines and prophylactic antibiotics. In another developing country, Jamaica, the mortality in children has fallen dramatically with these few, inexpensive interventions. More explanation is needed, such as the causes of death, given that these interventions are available in Brazil and hydroxyurea use (50%,29%) is relatively high for non non-developed country.

Are there publications from Brazil about economic burden of other diseases? This would aid in putting these results into context.

The other estimates of mortality rates should be discussed.

Reviewer #2: In this article, Carolina Bueno and colleagues estimate the monetary burden of Sickle Cell Disease (SCD) in Brazil and its effect and burden on the overall Brazilian Health care. The authors build a model using a one-year time horizon and considering direct medical costs and also indirect costs related to morbidity and mortality. The study concluded that burden for SCD care is at 1.6 billion Brazilian reals (approximately $300 million) pear year. The presented data also point to early childhood mortality, not seen in US, for example, and vaso-occlusive crises as the most frequent complication of SCD. Overall this is an interesting study, which is the first to estimate the burden of SCD in Brazil. The study provides a wealth of useful data and will be helpful for future public policy making decisions in Brazil. One major drawback is that the model is not well described (see detailed comments below). Also the authors need to fix numerous minor errors and provide additional citations and expand the discussion.

Major critiques:

1. Please provide more clear description of the model.

2. Please, provide more information about direct medical cost estimation. What is bottom-up strategy? Micro-costing method? Are there reference papers describing this methodology and strategy?

3. Figure 3 needs to have axis labels. It is unclear if this graph displays the estimated number of deaths per age group. Were these values per year or for the whole study, since data analysis was performed from 2008 to 2018? Please, clarify this in the text and also in the figure legend.

4. Authors should be more specific regarding which criteria and how medical experts validated complicated estimates.

5. Discussion, paragraph 9: you can expand the discussion and add references regarding deterministic sensitivity analysis and the influence of SCD prevalence, annual income and SCD base utility in disease burden.

Minor changes:

1. Page numbers need to be added.

2. Introduction, lane 5: “haemoglobin SS” – please change to “hemoglobin SS”. Also change throughout the manuscript.

3. Introduction, lane 7: “it is globally observed due to population migration.” – Slave trade is not exactly population migration, I suggest rephrasing it to clarify.

4. Introduction, lane 8: you may consider abbreviating sickle cell anemia as SCA.

5. Page 11, second paragraph: most of the available studies report only costs among hospitalized patients. However, Shah and coworkers (2020) reported the impact of vaso-occlusive crises in disease burden in non-hospitalized patients. Saraf and coworkers (2020) also demonstrated the cost impact of disease in a more specific SCD population. These manuscripts might be interesting to add to the discussion.

6. Tables 1, 2 and 3: References do not specify if they refer to children or adults. Please, clarify.

7. Table 3, please identify the acronym “SWAY”.

8. Page 12: “the main health states related to SCD (Error! Reference source not found.)” – please correct references.

9. Page 14, lane 6 “ data was defined” - “ data were defined.

10. Page 20, lane 25 “ estimated total DALY lost in 2018” should be “estimated total DALY loss in 2018”

6. PLOS authors have the option to publish the peer review history of their article (what does this mean?). If published, this will include your full peer review and any attached files.

Reviewer #1: No

Reviewer #2: No

---

## [Author Response · Author response to Decision Letter 0]

20 May 2022

The response to reviewers letter is attached in the new documents with all the answers required by the reviewers.

---

## [Editor Report · Decision Letter 1]

26 May 2022

Economic burden of sickle cell disease in Brazil

PONE-D-22-01357R1

Dear Dr.Carolina,

We’re pleased to inform you that your manuscript has been judged scientifically suitable for publication and will be formally accepted for publication once it meets all outstanding technical requirements.

Kind regards,

Mohamed A Yassin, MD

Academic Editor

PLOS ONE

Additional Editor Comments (optional):

The manuscript can be published in its current form
---

## [Editor Report · Acceptance letter]

2 Jun 2022

PONE-D-22-01357R1 

Economic burden of sickle cell disease in Brazil. 

Dear Dr. Bueno:

I'm pleased to inform you that your manuscript has been deemed suitable for publication in PLOS ONE. Congratulations! Your manuscript is now with our production department. 

Kind regards, 

on behalf of

Dr. Mohamed A Yassin 

Academic Editor

PLOS ONE